# Unifying Local and Global Knowledge: Empowering Large Language Models as Political Experts with Knowledge Graphs

## ABSTRACT

Large Language Models (LLMs) have revolutionized solutions for general natural language processing (NLP) tasks. However, deploying these models in specific domains still confronts challenges like hallucination. While existing knowledge graph retrieval-based approaches offer partial solutions, they can not be well adapted to the political domain. On the one hand, existing generic knowledge graphs lack vital political context, hindering deductions for practical tasks. On the other hand, the nature of political questions often renders the direct facts elusive, necessitating deeper aggregation and comprehension of retrieved evidence. To address these challenges, we present a **P**olitical **E**xperts through Knowledge **G**raph Integration (**PEG**) framework. PEG entails the creation and utilization of a multi-view political knowledge graph (MVPKG), which integrates U.S. legislative, election, and diplomatic data, as well as conceptual knowledge from Wikidata. With MVPKG as its foundation, PEG enhances existing methods through knowledge acquisition, aggregation, and injection. This process begins with refining evidence through semantic filtering, followed by its aggregation into global knowledge via implicit or explicit methods. The integrated knowledge is then employed in LLMs through prompts. Experiments on three real-world datasets across diverse LLMs reaffirm PEG's superiority in tackling political modeling tasks.

## CCS CONCEPTS

• **Computing methodologies** → **Natural language processing**; • **Human-centered computing** → *Collaborative and social computing*.

## KEYWORDS

large language models, knowledge graph, political science

**ACM Reference Format:**
Anonymous Author(s). 2023. Unifying Local and Global Knowledge: Empowering Large Language Models as Political Experts with Knowledge Graphs. In *Proceedings of The Web Conference 2024 (WWW2024)*. ACM, New York, NY, USA, 12 pages. https://doi.org/XXXXXXX.XXXXXXX

## 1 INTRODUCTION

Large Language Models (LLMs) have exhibited impressive ability to tackle a wide range of tasks. As the scale of LLMs continues to expand, they possess the ability to answer questions based

Figure 1: (a) An illustration of GPT-3.5 [40] answering a roll call vote prediction question. It declined to answer due to outdated internal knowledge limitations. (b) Prompting LLMs with generic knowledge from Wikidata [49]. Existing solutions [3] cannot find the direct fact and generate the wrong answer. (c) Our PEG framework retrieves local evidence from our political domain KG, derives global knowledge, and unifies both knowledge, to generate the correct answer.

on their inherent knowledge, eliminating the need for additional fine-tuning [6, 32]. Nevertheless, when deployed in specific domains, these models still encounter certain challenges. Since intrinsic knowledge can be incomplete and outdated, LLMs may refuse to respond to a question or produce factually incorrect answers, leading to the well-known *hallucination* phenomenon [45]. This issue is particularly evident in the political domain, as shown in Figure 1(a), where LLMs struggle to perform tasks like political actor modeling and opinion mining without external knowledge, such as social context and expert knowledge.

To compensate for the knowledge gap of LLMs, a line of research proposes retrieval-based methods to augment LLMs via contextually relevant external knowledge. Early work [17, 20, 28, 46] utilize documents as sources of knowledge. Compared to documents, knowledge graphs (KGs) consisting of triples, *i.e.*, {(head entity, relation, tail entity)}, provide brief and explicit structural

knowledge and explainable reasoning paths [54]. An additional advantage of KGs lies in their adaptability and expansibility, allowing for seamless modifications and additions. Considering this, some efforts [3, 47, 54] were made recently, to prompt LLMs to answer questions that can be resolved by referencing KGs. This is accomplished by providing LLMs with plain text, reformatted paths, or mindmaps that contain basic triples extracted from KGs.

Despite the remarkable achievements in general knowledge graph question answering (KGQA) tasks, existing approaches that enhance LLMs with KGs fall short in addressing challenges specific to the political domain. This limitation stems from several key factors: (1) **Knowledge-Task Mismatch.** Existing knowledge graphs, such as Wikidata [49], Freebase [4], and YAGO [43], primarily contain domain-specific political information such as politicians' names and nationalities. However, this information is inadequate for capturing the opinions and stances of politicians on specific policies - a critical aspect that attracts more attention in political domain question answering. The mere inclusion of this basic knowledge into LLMs is insufficient for generating accurate responses due to the inherent mismatch between the available knowledge and the depth of inquiry within the political domain; (2) **Ineffective Direct Fact Retrieval.** Even if we enrich the current knowledge graphs with politicians' historical opinions about specific policies, existing methods for direct fact retrieval [2] or path reasoning [47] may still encounter difficulties when applied to prediction tasks like vote prediction and event prediction, as the future fact can not be directly retrieved in KGs. As shown in Figure 1(b), existing KG-enhanced LLM frameworks like KAPING [3] is not able to produce accurate answers because it struggles to locate the relevant facts or construct the expected reasoning paths; (3) **Lack of Semantic Understanding.** The current KG-enhanced LLM approaches typically focus on presenting local evidence and path links while neglecting the nuanced semantic relationships between pieces of evidence and their derived high-level contextual clues, resulting in an incomplete comprehension of the acquired knowledge.

In light of these challenges, we propose to enhance Large Language Models as **P**olitical **E**xperts through Knowledge **G**raph Integration (**PEG**). This framework leverages political domain knowledge to incorporate background information and augment LLMs in computational political tasks, comprising two key components:

First, to address the knowledge-task mismatch issue, we start with constructing a **multi-view political knowledge graph**, covering factual knowledge pertaining to U.S. politics, including legislation, election, and diplomatic events. This knowledge graph supplements generic conceptual knowledge by providing a tailored foundation for political expertise.

Second, based on this KG, we augment LLMs' inference by **knowledge acquisition, aggregation, and injection**. In particular, for each question, we extract relevant entities and explore their associated facts as candidate evidence. We then filter the evidence according to their semantic similarity to the question to reduce noise. This process intends to effectively retrieve relevant knowledge. After acquiring local evidence, we proceed to aggregate the local evidence into global knowledge, either implicitly through embedding techniques or explicitly making use of the strong summarization and reasoning capabilities of LLMs. Finally, we incorporate both the local evidence and global knowledge along

**Table 1: Examples of conceptual knowledge and factual knowledge in the MVPKG.**

| Conceptual Knowledge Examples | Factual Knowledge Examples |
|---|---|
| (Donald Trump, occupation, real estate entrepreneur) | (Andre Carson, sponsor bill, Patient Advocate Tracker Act…) |
| (Donald Trump, member of political party, Republican Party) | (Andre Carson, vote yea, Women's Health Protection Act of 2021…) |
| (Donald Trump, country of citizenship, United States of America) | (2020 United States presidential election in Colorado, successful candidate, Joe Biden) |
| (Donald Trump, owner of, Kingdom 5KR) | (Yemen, Host a visit, Barack Obama, 01/01/2010) |

with the question through pre-defined prompt templates, to guide LLMs in producing answers grounded in the provided knowledge with semantic understanding. As shown in Figure 1, **PEG** harnesses the cognitive capabilities and reasoning prowess of LLMs to consolidate localized evidence into a comprehensive body of contextual knowledge. This approach enables LLMs to deliver answers with greater depth, as exemplified in the case of attitudes on abortion issues. Here, LLMs can navigate a succinct reasoning path, *{H.R.26->against->abortion rights, Andre Carson->support->abortion rights}=>Andre Carson->against->H.R.26*, thereby eliminating the need for complex reasoning based solely on local evidence.

Our main contributions can be summarized as follows:

- To enhance large language models as political experts, we particularly construct a domain-specific political knowledge graph involving contemporary U.S. political facts of multiple perspectives, which consists of 116,176 entities, 602 relations and 1,857,410 triples.
- We introduce a novel approach for mining high-level knowledge from localized facts, thus addressing situations where direct answers within knowledge graphs prove elusive. We provide both implicit and explicit implementations for different types of LLMs.
- We have conducted comprehensive experiments on various real-world datasets and across different LLMs. Our proposed methodology consistently demonstrates competitive performance in comparison to established baselines. Furthermore, a thorough analysis affirms the superior performance and interpretability of our approach.

## 2 MVPKG: A MULTI-VIEW POLITICAL KNOWLEDGE GRAPH

While existing generic knowledge graphs such as Wikidata KG [49], FreeBase [4] and YAGO [43] have proven valuable in a range of NLP tasks, their utility is largely confined to addressing basic demographic queries. These knowledge graphs, however, lack the capability to effectively support complex tasks related to political actor modeling and argumentative reasoning in politics-related tasks. To our knowledge, merely a single attempt [12] has been made to construct a politics-related knowledge graph, but it concentrates on the congressional employment status, neglecting the broader spectrum of behavior-related insights. Because of these limitations, we propose to construct a knowledge graph that is both **political domain-specific**, with a keen emphasis on political knowledge, and **multi-view**, covering diverse situations within U.S. politics. Generally, we start with extracting U.S. political conceptual knowledge in Wikidata KG [49] and extend it by incorporating

factual knowledge, thereby ensuring a more comprehensive and nuanced understanding of political dynamics. Table 1 shows several examples of conceptual knowledge and factual knowledge in MVPKG.

## 2.1 Political Conceptual Knowledge

To begin with, we select entities within U.S. politics, including the President and Cabinet members, Congressional legislators, Governors, and various government offices, as the seed entities. Subsequently, we manually retrieve their unique QID identifiers from Wikidata [49] and proceed to query all 1-hop facts to form a KG subset, which is named **baseKG**. This process ends with 71,646 entities, 368 relations and 103,174 triples.

## 2.2 Political Factual Knowledge

Based on baseKG, we further expand our knowledge graph with more factual knowledge. Our objective is to ensure that the knowledge graph is multi-view, effectively covering key facets of U.S. politics, including legislation, elections, and diplomatic events spanning the past few decades. To achieve this, we have sourced data from a variety of resources and structured it for integration into the knowledge graph. Specifically, we use Legiscan API[1] to obtain legislative information including bills, sponsorship details and voting records of legislators. Data related to elections has been primarily collected from public sources such as Ballotpedia,[2] Wikipedia[3] and Cha, Kuriwaki, and Snyder [7]. This data contains election results for various offices including the President, Congressional House, Congressional Senate, Governor, State Houses, State Senate, and Mayors. For diplomatic events, records of interactions involving socio-political actors have been extracted, with a specific focus on those related to U.S. politics [5].

Most of the data crawled is structured or semi-structured. We organize the original data through certain rules so that all facts are expressed in the form of (subject, predicate, object) to conform to the storage format of the knowledge graph. Note that event data includes timestamps, which do not impact subsequent usage, as they can be treated as part of the factual context. As shown in Table 1, these historical factual facts provide clues for understanding political actors and events.

After acquiring subgraphs from each of these perspectives, we employ the existing entity linking tool [1, 18] to align entities with those already present in baseKG. Additionally, we further use BERT [24] to encode entities, merge entities with a similarity exceeding 0.95 to existing entities, and add the unmatched entities into baseKG. As a result, we obtain a comprehensive knowledge graph - **MVPKG**, which is composed of 116,176 entities, 602 relations and 1,857,410 triples.

## 3 EMPOWERING LARGE LANGUAGE MODELS WITH POLITICAL KNOWLEDGE

Figure 2 illustrates the overview of our proposed framework. We first acquire local evidence by entity-centric exploration and semantic-based filtering. Following this, we aggregate the local evidence to

[1] https://legiscan.com/
[2] https://ballotpedia.org/
[3] https://www.wikipedia.org/

derive the global knowledge in a hidden space. To enhance interpretability and adapt to more LLMs, we also offer an explicit solution to express global knowledge in natural language. Finally, we perform prompt engineering to inject both local and global knowledge into LLMs' understanding and reasoning process.

## 3.1 Knowledge Acquisition

Given a question $q$, our goal is to retrieve a sub-graph $\mathcal{G}_q$ consisting of a set of fact triples $\{(e_h, r, e_t)\}$ from an external KG $\mathcal{G} = \{(e_h, r, e_t)\} \in \mathcal{E} \times \mathcal{R} \times \mathcal{E}$, where $\mathcal{E}$ and $\mathcal{R}$ are sets of entities and relations, and $e_h$, $r$ and $e_t$ stand for the head entity, relation and tail entity, respectively.

*3.1.1 Entity-centric Evidence Exploration.* Since some key entities such as individuals and organizations are crucial in political scenarios, our initial step is to extract entities mentioned in the given question. Entity matching is implemented by existing entity-linking techniques [1, 18, 29]. This process yields an entity set $\mathcal{E}_q$ for exploration of evidence. We regard all the 1-hop fact triples associated with entities in $\mathcal{E}_q$ as the candidate fact triples, forming the candidate subgraph in Figure 2.

*3.1.2 Semantic-based Evidence Filter.* To simplify the process, one might consider injecting all candidate evidence related to entities directly into LLMs. However, this method suffers from limitations on input length and the potential for introducing noise, given the substantial number of associated triples, many of which might not be relevant to the question at hand. To address this challenge, we propose to further filter the evidence based on semantics. Firstly, we verbalize each fact triple which involves converting symbolic triples into text strings. We achieve this by concatenating the names of the head entity, relation and tail entity.

After verbalization, each fact triple can be regarded as a document and we can apply existing dense retrieval patterns [23, 56] to retrieve relevant evidence based on embedding similarities. To elaborate, we use the same sentence encoder to embed both question and fact triples and compute their similarity. In this way, for each triple, we can define a retrieval score as the inner product between the embeddings of the given question $q$ and the candidate triple $t$, as follows:

$$s(t \mid q) \propto \exp\left(d(t)^\top d(q)\right) \tag{1}$$

where $d$ is the embedding function. Subsequently, we only reserve top-$K$ fact triples $\mathcal{G}_l = \{(e_h, r, e_t)\}$ as our local evidence, where $K$ is a pre-defined hyper-parameter.

## 3.2 Knowledge Aggregation

After knowledge acquisition, most existing work [3, 47, 54] often directly prompts LLMs with plain text or reformatted paths. However, these approaches often overlook the semantic relationships that underlie the facts within the knowledge graph. In this section, we describe how we provide LLMs with more comprehensive knowledge. This enhancement is achieved by further aggregating local evidence to form global and more generalized knowledge compared to the fine-grained evidence, to better deal with the situations where direct facts are not readily matched, and relying only on local evidence proves insufficient.

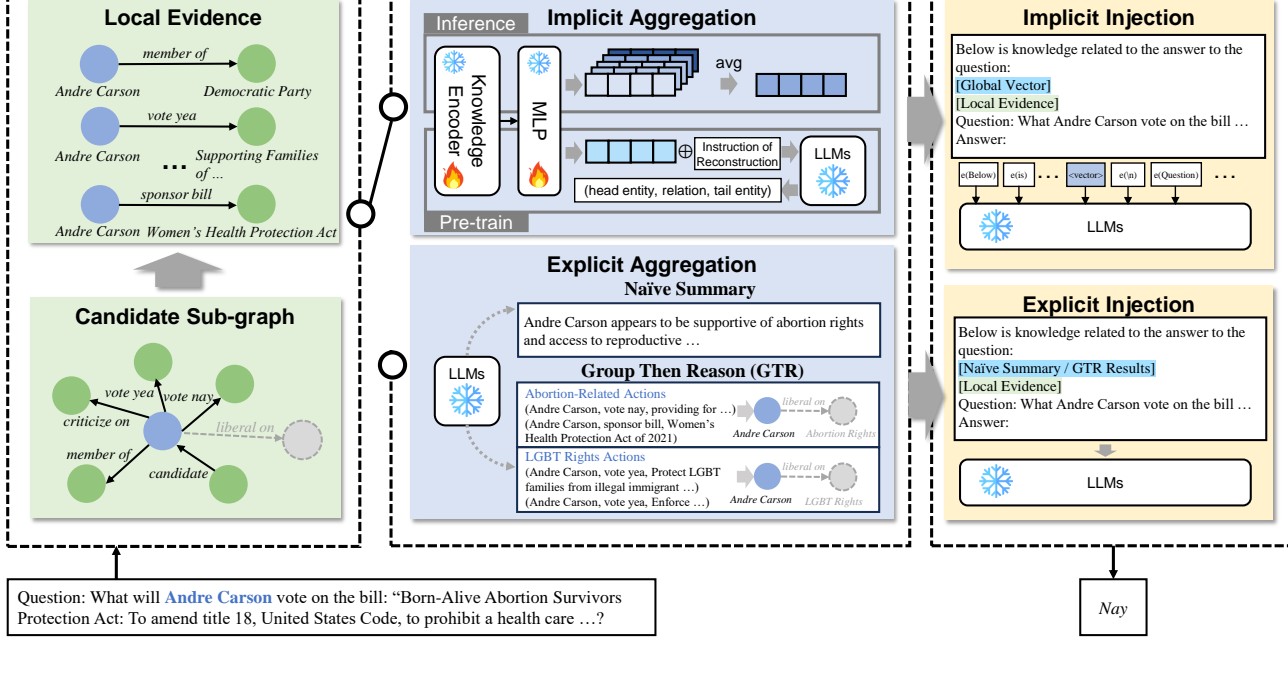

**Figure 2: After constructing the multi-view political knowledge graph (MVPKG), we empower large language models with political knowledge through knowledge acquisition, aggregation, and injection. For knowledge aggregation, we implement two options: (1) implicit aggregation with embedding techniques, and (2) explicit aggregation with natural language.**

*3.2.1 Implicit Aggregation.* Intuitively, we can aggregate the semantic information of retrieved facts in the embedding space. Here, we introduce another language model as a global knowledge encoder. Concretely, we use the knowledge encoder to encode the retrieved fact triples separately and average the sentence embedding as the global topic vector. This topic vector is further processed through a multilayer perceptron (MLP), to ensure alignment with the semantic space of LLMs, as follows:

$$v = \frac{1}{K} \sum_{k=1}^{K} MLP(Encoder(t_k)) \quad (2)$$

where $t_k$ is the $k$-th triple after evidence filtering in Section 3.1.2.

This topic vector $v$ serves as a soft prompt to facilitate LLMs in answering the questions effectively. However, it is worth noting that a knowledge encoder and an MLP with random initialization or general pre-training may perform poorly in transforming information from the space of the original knowledge encoder to that of the target LLMs. This is due to a lack of specific training for this purpose. Thus, we propose a *fact reconstruction* task to pre-train the knowledge encoder and MLP components.

Specifically, we use the knowledge encoder to acquire the sentence embedding of each single fact triple and prompt LLMs to reconstruct the texts of the given triple based on the embedding. This task enables the knowledge encoder and MLP to express fact-related information in LLMs' space, without additional annotated data. In essence, we leverage signals from a fixed language model

to train the encoder, using a language modeling objective:

$$p(\mathbf{y} \mid \mathbf{x}, \mathbf{v}) = \prod_{t=1}^{T} p\left(y_t \mid \mathbf{x}, \mathbf{v}, \mathbf{y}_{0:t-1}\right) \quad (3)$$

where $\mathbf{y} = [y_1, ..., y_T]$ is the output response, *i.e.*, the text of input triple, $\mathbf{x} = [x_1, ..., x_N]$ is the prompt instructing LLMs to reconstruct the triple based on the given vector $\mathbf{v}$. Note that this process can be applied to any set of triples, and once the knowledge encoder is trained, it can be adapted to the frozen LLMs for inference directly.

*3.2.2 Explicit Aggregation.* Although aggregating vectors in a hidden space is straightforward, explaining what knowledge the vectors actually represent is not trivial. Relying on the strong reasoning and generation ability of large language models, we aggregate the facts in an explicit manner, where the derived knowledge is expressed in natural language, to provide a global view of the local evidence. This method is more flexible since it can be applied to black-box LLMs like ChatGPT [40], where inputs in the form of vectors are often not acceptable. The simplest way to achieve this goal is to prompt LLMs to reason and summarize the given evidence in natural language $\mathbf{w} = [w_1, ..., w_i]$, similar to the principles behind many prior studies [25, 36, 53]. We achieve this by instructing LLMs with the question, *"What can you infer from the following facts?"* This variant is named **Naïve Summary**.

While the idea is intuitive, the dispersion of evidence can result in ambiguous and cluttered summaries. Often, these summaries may merely consist of repetitions or abstracts of individual facts.

To address this issue, we introduce a **Group Then Reason (GTR)** strategy. Specifically, we add the instruction to prompt LLMs to divide the evidence into several groups $g_1, ..., g_M$ according to the topical information, and summarize the reasoning result of each group into new fact triples $\mathcal{G}_h = \{(e_h, r, e_t)\}$. Symbolically, this approach generates new pseudo-entities that convey topical information and their relationships with existing entities. This enhances the potential for answering questions that may not have direct matches in the knowledge graph. By organizing evidence into topical categories, it becomes easier for LLMs to reason about implicit knowledge hidden within the evidence, such as a politician's ideology or attitudes towards subjects within the same category.

## 3.3 Knowledge Injection

Once we have collected the local evidence and aggregated global view, the next step is to inject the knowledge, allowing LLMs to provide answers that are rooted in the associated external knowledge. For explicit aggregation, we integrate the verbalized local evidence $\mathcal{G}_l$ and the aggregated result $w$ or $\mathcal{G}_h$ using a pre-defined instruction template. This prompt is then placed at the beginning of the input question $q$, to stimulate LLMs to generate answers conditioned on the provided knowledge. The process can be formalized as $p(y \mid [\mathcal{G}_h, \mathcal{G}_l, q])$, where $[\cdot]$ denotes concatenation.

For implicit aggregation, inspired by prompt tuning [27] and P-tuning [34], we regard the global vector as a soft token and concatenate it with token embeddings derived from the verbalized local evidence $\mathcal{G}_l$ and the question. The resulting sequence of token embeddings is then fed into the transformer layers of the LLMs.

Note that we are following the zero-shot setting, where we do not possess any labeled samples or train models. This differs from supervised learning [2, 22], where models are trained with a set of (question, answer) pairs or (question, ground-truth facts) pairs.

## 4 EXPERIMENTS

To showcase that MVPKG and the proposed framework for knowledge integration can generally assist various tasks in the political domain, we conduct comprehensive experiments.

## 4.1 Experiment Settings

*4.1.1 Tasks and Datasets.* We employ three datasets representing various political scenarios for the assessment. **RCVP** [38] is a congressional roll-call vote prediction dataset. We further collect 7,927 voting records in 2023 from Legiscan for evaluation. The historical records are integrated into KGs to align with the time-based setting proposed in Mou et al.. As per the conventions outlined in prior studies [38, 39], we report the macro F1 score for this binary classification task. **ICEWS** [5] contains political diplomatic events where we reserve 9,322 samples from 01/01/2023 to 04/10/2023 for evaluation. Following Zhu et al., we formulate the task as predicting either the subject or the object of each event. For each question, we randomly sample three negative entities in the same category with the ground-truth answer to form a multiple-choice setting. Accuracy is reported. **StaId** represents a statement identification task curated in this paper. We sample 4,000 tweets from Mou et al. and create questions that revolve around determining whether a

given statement on a specific issue was posted by a particular politician. This task assesses the capabilities of LLMs to comprehend politicians' attitudes on various issues. Macro F1 is reported for this binary classification task.

*4.1.2 Compared Methods.*

*Baselines without External Knowledge.*
- **Vanilla**, *i.e.*, providing questions directly to LLMs.
- **GKP** [31] extracts knowledge from LLMs themselves and then prompts LLMs with the generated knowledge.

*Baselines with Local Evidence Only.*
- **KAPING** [3] retrieves the knowledge based on similarity and prompts the textual triples to LLMs.
- **MindMap**$_{route}$ [54] clusters the retrieved triples into structured pathways like *2020 U.S. state House of Representatives elections in District 75 of Iowa->candidate->(Ruby Bodeker, Thomas Gerhold)*, which is subsequently prompted to LLMs.
- **MindMap**$_{lang}$ [54] prompts LLMs to describe the evidence route in natural language and leverage the generated content for further prompting.
- **MindMap** [54] prompts LLMs to answer the questions and meanwhile describe the evidence route and construct a decision tree-like mindmap.

*Our Methods.*
- **PEG**$_{imp}$, our method with implicit aggregation.
- **PEG**$_{exp\_sum}$, our method with explicit aggregation, where the global knowledge is generated through Naive Summary.
- **PEG**$_{exp\_GTR}$, our method with explicit aggregation, where the global knowledge is generated through GTR.

*4.1.3 Implementation Details.* We use several LLMs to verify the effectiveness of our framework, including LLAMA2-7B-CHAT [48], VICUNA-7B-v1.1 [60] and GPT-3.5-TURBO [40]. We use ReFinED [1] for entity linking and a document-pretrained distillbert[4] as the retriever for semantic filtering. We use the KAPING [22] methods to arrange local evidence. Top-10 facts are reserved for knowledge graph integration. When evaluating white-box LLMs including LLAMA2 and VICUNA, we follow Li et al. to concatenate each candidate answer with the input and compare the language modeling likelihood to determine the answer for a stable evaluation. When it comes to black-box LLMs such as GPT-3.5, we evaluate based on the generated results since likelihood is not available. Unless otherwise specified, all KG-enhanced methods use knowledge sourced from MVPKG. More details can be found in Appendix A.1.

## 4.2 Experiment Results

*4.2.1 Main Results.* Table 2 presents the primary results across various tasks and white-box language models. In general, our proposed methods *consistently* outperform other baselines. It is important to note that the generated knowledge model (GKP) is not significantly superior to the vanilla knowledge-free model, since in most cases of our political tasks, LLMs can not generate relevant and accurate knowledge about future facts, limiting their assistance in improving answers. Conversely, KG-augmented methods clearly outperform

---

[4]https://huggingface.co/sentence-transformers/msmarco-distilbert-base-v3

**Table 2: Main results of white-box large language models. The best scores are emphasized in bold.**

| Methods | RCVP | | ICEWS | | StaId | |
|---|---|---|---|---|---|---|
| | Llama2 | Vicuna | Llama2 | Vicuna | Llama2 | Vicuna |
| Vanilla | 40.07 | 37.17 | 23.98 | 22.88 | 57.10 | 49.57 |
| GKP [31] | 42.95 | 35.71 | 29.40 | 24.47 | 56.56 | 52.73 |
| KAPING [3] | 44.66 | 42.92 | 39.80 | 36.06 | 53.67 | 53.84 |
| MindMap$_{route}$ [54] | 43.41 | 43.73 | 40.07 | 36.12 | 52.67 | 53.85 |
| MindMap$_{lang}$ [54] | 44.67 | 42.44 | 37.87 | 33.94 | 53.03 | 45.84 |
| MindMap [54] | 43.45 | 40.24 | 33.38 | 29.95 | 55.42 | 55.27 |
| PEG$_{imp}$ | **52.56** | **44.56** | 37.47 | 34.25 | 56.36 | 52.07 |
| PEG$_{exp\_sum}$ | 47.77 | 42.75 | 38.60 | 35.06 | **58.67** | 53.67 |
| PEG$_{exp\_GTR}$ | 47.49 | 44.16 | **40.10** | **36.51** | 55.11 | **54.76** |

**Table 3: Main results of GPT-3.5. The best scores are emphasized in bold. Note that due to the constraint of the input format and the inherent inability to predict the future, we do not include results of PEG$_{imp}$ and the vanilla baseline.**

| Methods | RCVP | ICEWS | StaId |
|---|---|---|---|
| GKP [31] | 20.43 | 15.40 | 45.32 |
| KAPING [3] | 37.83 | 20.80 | 36.99 |
| MindMap$_{route}$ [54] | 32.60 | 28.00 | 35.03 |
| MindMap$_{lang}$ [54] | 38.57 | 28.40 | 42.19 |
| MindMap [54] | 38.72 | 25.60 | 23.38 |
| PEG$_{exp\_sum}$ | 40.62 | 26.40 | 42.12 |
| PEG$_{exp\_GTR}$ | **41.21** | **28.60** | **46.21** |

the naive models, underscoring the value of the knowledge contained within MVPKG for addressing political tasks. Among these methods, our methods demonstrate distinct advantages, suggesting that simply arranging local facts or describing the structural relations among facts is **insufficient** for tackling complex tasks like vote prediction. Integrating global knowledge can lead to **substantial enhancements**. Additionally, building a mindmap proves to be a challenging endeavor, especially for smaller-scale LLMs. Consequently, MindMap falls short of achieving superior results compared to its simpler counterparts, MindMap$_{route}$ and MindMap$_{lang}$. In contrast, summarizing and reasoning about the local evidence is not only more feasible but also cost-effective.

Through comparison, it is evident that various datasets and tasks necessitate distinct aggregation strategies. In the case of RCVP, **PEG** with implicit aggregation proves most effective, primarily because the retrieved facts predominantly consist of historical voting records with lengthy text, and the explicitly summarized content may include irrelevant information. Conversely, for ICEWS and StaId, **PEG** with explicit aggregation exhibits a slight advantage. This difference could be attributed to the retrieved facts being less concentrated compared to the voting records in RCVP, making the averaging of facts in the latent space potentially less meaningful.

Table 3 provides the results of GPT-3.5. Relying on the rich internal knowledge of GPT-3.5, the advantage of the GKP method becomes evident in the StaId dataset. However, it still struggles to handle tasks like RCVP. Comparatively, our methods show *consistent* superiority in assisting this black-box LLM across diverse tasks.

**Table 4: Results of adding global knowledge to different patterns of local evidence.**

| Evidence Format | Integration Method | RCVP | ICEWS | StaId |
|---|---|---|---|---|
| KAPING | w/o global knowledge | 44.66 | 39.80 | 53.67 |
| | PEG$_{imp}$ | **52.56** | 37.47 | 56.36 |
| | PEG$_{exp\_sum}$ | 47.77 | 38.60 | **58.67** |
| | PEG$_{exp\_GTR}$ | 47.49 | **40.10** | 55.11 |
| Mindmap$_{route}$ | w/o global knowledge | 43.41 | 40.07 | 52.67 |
| | PEG$_{imp}$ | **48.67** | 39.84 | **55.63** |
| | PEG$_{exp\_sum}$ | 47.48 | 40.31 | 53.49 |
| | PEG$_{exp\_GTR}$ | 48.11 | **40.37** | 53.01 |
| Mindmap$_{lang}$ | w/o global knowledge | 44.67 | 37.87 | 53.03 |
| | PEG$_{imp}$ | 45.03 | 38.17 | 53.90 |
| | PEG$_{exp\_sum}$ | 45.46 | **39.68** | **54.52** |
| | PEG$_{exp\_GTR}$ | **45.65** | 39.63 | 49.29 |

The improvement brought by PEG$_{exp\_GTR}$ is more pronounced for GPT-3.5 than for Llama2 and Vicuna, as GPT-3.5 poses stronger capabilities to group and summarize local information.

*4.2.2 Effectiveness of MVPKG.* To demonstrate the effectiveness of our MVPKG, we compare it with other generic knowledge graphs including Wikidata KG [49], *i.e.*, baseKG in this paper, and YAGO [43] and political knowledge graph KGAP proposed in Feng et al.. Figure 3 shows the performance paired with different KGs on Llama2. The results suggest that domain knowledge graphs including KGAP and our MVPKG are more practical than generic KGs in understanding the political actors and events since they assist models in learning from related factual history. Moreover, our knowledge graph is more comprehensive than KGAP, encompassing various aspects of U.S. politics. Consequently, it exhibits strengths across diverse datasets. Surprisingly, Wikidata demonstrates competitive performance on the StaId dataset. This could be attributed to the fact that some statements' content is partially linked to fundamental attributes of politicians, such as party affiliation and home state.

*4.2.3 Effectiveness of Global Knowledge.* In order to illustrate the effectiveness of global knowledge, we test our **PEG** framework with different patterns of local evidence, *i.e.*, KAPING, MindMap$_{route}$ and MindMap$_{lang}$. Table 4 shows the results of this ablation study. Overall, global knowledge works across three different forms of local evidence. PEG$_{exp\_sum}$ exhibits the most consistent performance, since the global knowledge in a format of natural languages is the most straightforward for a language model to comprehend. While for PEG$_{exp\_GTR}$ on Mindmap$_{route}$ and Mindmap$_{lang}$, LLMs need to understand at least two forms of knowledge, *i.e.*, local evidence in forms of path or language and global knowledge in forms of triple. Thus, the gain is unstable. In contrast, PEG$_{exp\_GTR}$ with KAPING clearly improves the KAPING baseline, since both local and global knowledge is expressed in fact triples.

In response to the challenge of limited domain knowledge comprehension, some researchers [9, 58] have turned to instruction tuning [35] using domain-specific data. We also compare our methods with this line of approaches that fine-tune the model to solve domain tasks. Experimental results reveal that instruction tuning

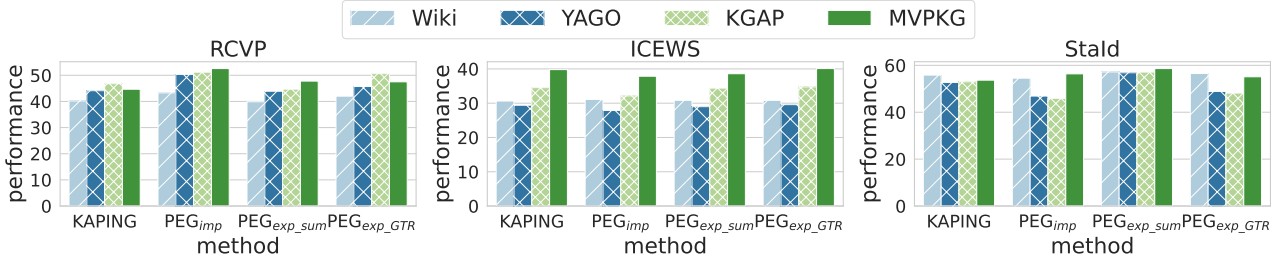

Figure 3: Performance of LLAMA2 when using different knowledge graphs for knowledge integration.

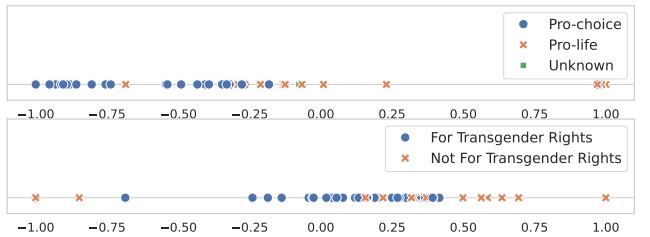

Figure 4: The PCA outputs of the global knowledge vectors corresponding to questions asking to predict Congressional members' votes on the Born-Alive Abortion Survivors Protection Act (upper) and the Protection of Women and Girls in Sports Act of 2023 (lower).

does not exhibit significant advantages over our external knowledge integration approach. Moreover, the augmentation of LLMs with knowledge graphs, instead of resorting to additional training, proves to be a more flexible and cost-effective solution, particularly in addressing the rapidly evolving political landscape. More details can be found in Appendix A.4.

## 5    FURTHER ANALYSIS

In this section, we conduct more experiments to implement an in-depth analysis of the PEG framework and global knowledge.

### 5.1    Explainability of Global Knowledge

Compared to explicit methods where reasoning results are expressed in natural language, the implicit method is much more difficult to explain. Intuitively, the aggregated vector contains information on one or more aspects related to the central entity with respect to the given question. To validate this, we take the RCVP dataset as an example for analysis by visualizing the PCA-transformed representation of the global knowledge vector. Figure 4 depicts the global knowledge vector after dimensionality reduction of questions asking to predict different Congressional members' votes on two bills respectively. We label each member's position on abortion issues (*Pro-choice* and *Pro-life* in Figure 4) and transgender issues (*For Transgender Rights* and *Not For Transgender Rights* in Figure 4) based on information provided by public websites.[5] [6] The visualization shows that samples with similar positions cluster

[5]https://justfacts.votesmart.org/
[6]https://www.ontheissues.org/

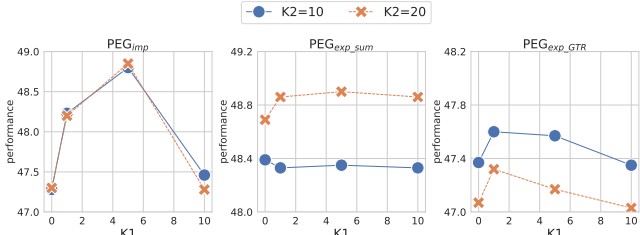

Figure 5: Performance with varying amount of knowledge, where we change the number of fact triples K1 for local evidence and K2 for global knowledge.

together. This indicates that the global knowledge vector aggregated by the knowledge encoder, using factual traces, can reflect the attitudes of key individuals on specific topics or issues. With this information, models can more easily deduce how members are likely to vote on emerging bills related to the same issue.

### 5.2    Impact of Amount of Knowledge

To explore the influence of the information load on both local evidence and global knowledge, we vary the number of facts used to form local evidence and global knowledge. Figure 5 displays the average results of three datasets when we use top $K1$ facts for local evidence and top $K2$ facts for global knowledge using different knowledge aggregation methods on LLAMA2. Firstly, the performance shows an increasing trend followed by a decrease as the quantity of local evidence changes in most cases. This trend occurs because including more clues can lead to improved results initially, but as the amount of provided facts increases, LLMs may become distracted by irrelevant fact triples, which hinders performance. This phenomenon is not significant for PEG$_{exp\_sum}$, possibly because it extracts more relevant information during the summary process. Also, it is noticeable that when relying solely on global knowledge (i.e., $K1 = 0$), these methods, particularly the explicit ones, can still produce competitive results. In contrast to general KGQA tasks, where the ground-truth answer triple is often expected in the retrieved results, our approach leverages both local evidence and global knowledge to provide relevant information rather than searching for specific answer triples. Furthermore, when $K2$ increases, the methods exhibit varying trends. PEG$_{exp\_sum}$ demonstrates an improvement in performance, whereas PEG$_{exp\_GTR}$ experiences a decline, mainly due to the drop in GTR performance

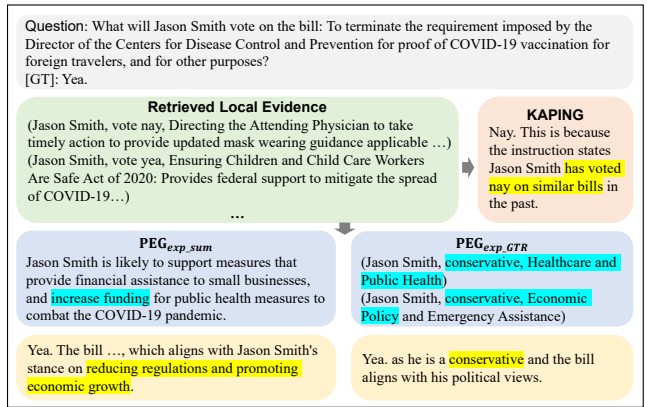

**Figure 6: A case from the RCVP dataset. Clues utilized by LLM inference and reasons for answers are highlighted.**

when more facts with longer texts are introduced. This indicates that the effectiveness of different knowledge aggregation methods can be influenced by the quantity of global knowledge used.

### 5.3 Case Study

Figure 6 shows a case from the RCVP dataset, where we further ask LLM why it outputs the answer. We can observe that even provided with relevant historical information, KAPING [3] cannot answer correctly. This is because bills of similar topics might express different leanings, but LLM does not fully comprehend the relationship between them. This again highlights the challenges in reasoning from such local evidence. In contrast, $PEG_{exp\_}$ and $PEG_{exp\_GTR}$ provide direct information about the politicians.

## 6 RELATED WORK

**Knowledge-augmented LLMs.** To mitigate the hallucination problem of large language models (LLMs), some recent works have leveraged external knowledge for LLM inference. Works represented by REALM [17], RAG [28] and Replug [46] first proposed to retrieve documents and augment LLMs with the retrieved unstructured corpora. Compared to documents, knowledge graphs, as another knowledge source alternative, are less constrained by the limited input length of LLMs and can express more explicit knowledge in a more compact form of fact triples. Previous research on KG-augmented language models or the synergization of KG and language models focused on designing additional modules and training objectives to incorporate knowledge in the pre-training, fine-tuning and inference stages of language models respectively [33, 42, 51, 52, 59]. With the recent progress of large language models, the research paradigm is shifting to prompting fixed LLMs without additional training. Baek et al. first proposed to retrieve fact triples from knowledge graphs based on similarity with a given question and prompt the triple texts to LLMs to handle the knowledge graph question answering questions. Guo et al. and Wang et al. have explored prompting LLMs for graph mining. To fill the gap in understanding structural information, Sun et al. iteratively retrieved triples and constructed reasoning paths

and Wen et al. further prompted LLMs to generate a decision tree-like mindmap to visualize the reasoning process and help better answer the questions. Although reporting positive results, existing methods focus on directly stacking all facts or focusing on path links between facts, while ignoring the semantic relationships between facts, *i.e.*, the global thematic information. Many of these methods are primarily designed and tested on general or medical domain Q&A datasets, where the correct answers are typically entities that can be directly matched within the corresponding knowledge graphs. Consequently, plain text answers or reasoning paths can provide meaningful and effective solutions. However, when these methods are applied to more complex tasks such as prediction, they may fall short since the direct facts or paths needed for accurate answers can no longer be readily found. In our study, we aim to address this challenge by experimenting with a simple yet effective approach - aggregating local facts to generate higher-level global knowledge that is both more directly relevant to the questions and capable of providing better signals for large language models.

**Political Actor Modeling and Opinion Mining.** Modeling the political actors and understanding their discourse is at the core of computational political science. There are a wide range of applications in various tasks such as roll call vote prediction [38], perspective detection [12] and frame detection [21, 41]. Early research focused on statistical analysis of roll call data to estimate the ideology of political actors. One of the most widely used methods for vote-based analysis is the Ideal Point Model [8], which reveals how the divisions among legislators reflect their partisan affiliations. Researchers have expanded upon this model by incorporating the texts of bills to enhance its accuracy [14, 26]. Recently, researchers have introduced more abundant social contextual information such as co-sponsorship network between legislators [57], hashtag network [38], relations of contributors [10], relations of stakeholders [11] and mention in documents [44]. Although these kinds of metadata have proven effective, collecting it in large quantities is expensive due to their complex and diverse data formats. Mou et al. proposed a unified scheme by injecting social information in the pre-training stage and using languages only to represent political actors and solve various downstream tasks. Considering large language models' strong ability in understanding and reasoning, we do not aim to train the models but construct a multi-view political knowledge graph where social information is expressed in a unified format of triple, and covering different scenes of U.S. politics whose source records are publicly available and continuously updated.

## 7 CONCLUSION

In this study, to address the challenge of insufficient political knowledge in large language models, we construct a comprehensive domain-specific political knowledge graph covering diverse facets of U.S. politics. Subsequently, we introduce the **P**olitical **E**xperts through Knowledge **G**raph Integration (**PEG**) framework to address the tasks of political actor modeling and opinion mining. Based on existing work, we unify the local and global knowledge using diverse methods, to alleviate the issue when direct answers can not be found in constructed knowledge graphs. Experiments across different datasets and LLMs demonstrate the effectiveness and explainability of the proposed approach.

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

# A EXPERIMENT DETAILS

## A.1 Hyperparameters

We present the hyperparameter settings in Table 6.

## A.2 Evaluation

For evaluation on white-box LLMs, *i.e.*, Llama2 and Vicuna, we follow Li et al. to implement a likelihood evaluation. Given knowledge context $k$, question $q$ and options $C = \{c^i\}_{i=1}^N$, the answer prediction can be determined by the generation likelihood predicted by the evaluated model:

$$\hat{c} = \arg\max_{c^i \in C} P_\theta \left( c^i \mid k, q \right) \quad (4)$$

where $P_\theta$ is parameterized by the causal-LLM.

For evaluation on black-box LLMs, *i.e.*, GPT-3.5, we provide options in prompt and ask the model to output its choice, since likelihood is not applicable in the API. And we further write regular

**Table 5: Performance on vanilla** Vicuna, **instruction-tuned** Vicuna **and KG-enhanced** Vicuna **(best PEG variant is reported).**

| Method | RCVP | ICEWS | StaId |
|--------|------|-------|-------|
| Vanilla | 37.17 | 22.88 | 49.57 |
| FT | 38.81 | 23.89 | 39.18 |
| PEG | **44.56** | **36.51** | **54.76** |

**Table 6: Hyperparameter settings.**

| Hyperparameter | Value |
|----------------|-------|
| Knowledge Encoder Training | |
| batch size | 1,024 |
| epochs | 5 |
| learning rate for encoder | 2e-5 |
| learning rate for MLP | 2e-4 |
| warmup ratio | 0.1 |
| encoder | distillbert [7] |
| Explicit Aggregation | |
| max tokens for summary generation | 128 |
| max tokens for GTR generation | 128 |
| do sample | False |
| temperature | 1 |
| num beams | 1 |
| LLM inference | |
| K | 10 |
| VectorDB for retrieval | FAISS |

expressions to match answers to deal with situations when options are not explicitly output.

## A.3 Prompts

We illustrate prompt examples in Figure 7 and Figure 8.

## A.4 Comparison with Domain Instruction-tuned LLMs

Facing the challenge of insufficient comprehension of domain knowledge, some researchers [9, 58] have employed instruction tuning [35] on domain-specific data. To compare directly providing external knowledge with internalizing knowledge through fine-tuning in the political domain, we curate an instruction dataset consisting of 28,187 samples, from 28 publicly available datasets across 11 tasks, such as stance detection [15, 37] and ideology detection [13, 55]. Subsequently, we use LoRA [19] to finetune Vicuna. As shown in Table 5, instruction tuning (FT) does not show obvious advantages over the external knowledge integration solution. Meanwhile, enhancing LLMs with KG instead of additional training appears to be a more adaptable and cost-effective solution for the rapidly evolving political landscape.

Received 20 February 2007; revised 12 March 2009; accepted 5 June 2009

**Figure 7: Prompt examples in PEG.**

| Prompt | Value |
|---|---|
| QA for RCVP, ICEWS, StaId | Below is knowledge related to the answer to the question:
{global knowledge}
{local evidence}

Question:
{question}

Answer: |
| Naïve summary generation | What can you infer from the following facts?

Facts:
{local evidence}

Inference: |
| GTR generation | Here are some fact triples in the form of (subject, predicate, object). Group the facts based on the topical information and summarize what you can infer from each group of facts into triples. Output the triples only.
Here is an example.

Input:
(Andre Carson, vote nay, Born-Alive Abortion Survivors Protection Act: To amend title 18, United States Code, to prohibit a health care practitioner from failing to exercise the proper degree of care in the case of a child who survives an abortion or attempted abortion.)
(Andre Carson, vote yea, Women's Health Protection Act of 2021: To protect a person's ability to determine whether to continue or end a pregnancy, and to protect a health care provider's ability to provide abortion services.)
(Andre Carson, vote nay, Providing for consideration of the bill (H.B. 4712) to amend title 18, United States Code, to prohibit a health care practitioner from failing to exercise the proper degree of care in the case of a child who survives an abortion or attempted abortion, and providing for proceedings during the period from January 22, 2018, through January 26, 2018)
(Andre Carson, vote yea, Supporting Families of the Fallen Act: A bill to amend title 38, United States Code, to increase automatic maximum coverage under the Servicemembers' Group Life Insurance program and the Veterans' Group Life Insurance program, and for other purposes.)
(Andre Carson, sponsor bill, Patient Advocate Tracker Act: To amend title 38, United States Code, to improve the ability of veterans to electronically submit complaints about the delivery of health care services by the Department of Veterans Affairs.)
(Andre Carson, sponsor bill, Protecting Our Kids Act: To amend title 18, United States Code, to provide for an increased age limit on the purchase of certain firearms, prevent gun trafficking, modernize the prohibition on untraceable firearms, encourage the safe storage of firearms, and for other purposes.)

Output:
(Andre Carson, liberal, abortion rights and abortion services)
(Andre Carson, support, increase veterans' group life insurance and healthcare services)
(Andre Carson, liberal, gun control for kids)

Try to output:
Input:
{local evidence}

Output: |

**Figure 8: Prompt for Mindmap generation we curate with reference to medical mindmap prompt in [54].**

| Prompt | Value |
|--------|-------|
| | Given some political knowledge information, answer the questions. Think step by step. Output three parts. |
| | Output1: The answer to the question. |
| | Output2: Show me inference process as a string about extract what knowledge from which Evidence, and in the end infer what result.
Transport the inference process into the following format:
Evidence number('entity name'->'relation name'->...)->Evidence number('entity name'->'relation name'->...)->Evidence number('entity name'->'relation name'->...)->...). |
| | Output3: Draw a decision tree. The entity or relation in single quotes in the inference process is added as a node with the source of evidence, which is followed by the entity in parentheses. |
| | Here is a sample: |
| | Knowledge:
(Andre Carson, vote nay, banning federal health coverage that includes abortion)
(Andre Carson, support, funding abortion avoids discrimination against poor women)
(Andre Carson, support, constitutional right to terminate pregnancy for health) |
| | Question:
What stance will Andre Carson take on baning anti-abortion limitations on abortion services? |
| | Output1:
Andre Carson will support baning anti-abortion limitations on abortion services. |
| Mindmap Generation | Output2:
Evidence 1('Andre Carson'->'vote nay'->'banning federal health coverage that includes abortion')->Evidence 2('banning federal health coverage that includes abortion'->'oppose'->'abortion')->Evidence 3('Andre Carson'->'support'->'funding abortion avoids discrimination against poor women')->Evidence 4('funding abortion avoids discrimination against poor women'->'support'->'abortion')->Evidence 5('Andre Carson'->'support'->'constitutional right to terminate pregnancy for health')->Evidence 6('constitutional right to terminate pregnancy for health'->'support'->'abortion'). |
| | Output3:
Andre Carson(Evidence 1)(Evidence 3)(Evidence 5)
├── vote nay(Evidence 1)
│   └── banning federal health coverage that includes abortion(Evidence 1)(Evidence 2)
│       └── oppose(Evidence 2)
│           └── abortion(Evidence 2)
├── support(Evidence 3)
│   └── funding abortion avoids discrimination against poor women(Evidence 3)(Evidence 4)
│       └── support(Evidence 4)
│           └── abortion(Evidence 4)
└── support(Evidence 5)
    └── constitutional right to terminate pregnancy for health(Evidence 5)(Evidence 6)
        └── support(Evidence 6)
            └── abortion(Evidence 6) |
| | Now try to output:
Knowledge:
{local evidence} |
| | Question:
{question} |

