# OpenReview forum: "Unifying Local and Global Knowledge: Empowering Large Language Models as Political Experts with Knowledge Graphs"
_ACM.org/TheWebConf/2024/Conference — TheWebConf24 Oral_

### Official Review · Reviewer_vVzz · 2023-10-28

**Novelty:** 5
**Technical Quality:** 4

**Review:**

This paper investigates the KBQA task in the political domain with LLM RAG. The paper first constructs a multi-view political knowledge graph (MVPKG) from Wikipedia as well as domain-specific source (Ballotpedia. Cha, Kuriwaki, Snyder) for political-related entities. Then a political experts through knowledge graph integration (PEG) framework is proposed. The experiment on three datasets with 3 different LLM shows the good performance of the proposed method.

Good:
- The study is comprehensive from KG construction to evaluation
- The experimental result shows that the domain-specific KG is very helpful for LLM to solve tasks in specific domains.

To improve:
- Did not find Quality Control or Error analysis about the MVPKG. I am not sure about the KG quality itself. Nor ablation study on general KB source is conducted. (line 566 shows only experiments on MVPKG are conducted) Hence, not sure about the gain is from MVPKG or methods.
- In line 596, no vanilla GPT-3-turbo is conducted due to "constraint of the input format and the inherent inability to predict the future" I did not quite understand it. The result should be presented for completeness even it may be very low.

**Questions:**

See to improve.

**Reviewer Confidence:**

3: The reviewer is confident but not certain that the evaluation is correct

**Scope:**

3: The work is somewhat relevant to the Web and to the track, and is of narrow interest to a sub-community

---

### Official Review · Reviewer_9UXj · 2023-11-23

**Novelty:** 5
**Technical Quality:** 7

**Review:**

The study presents a method to complement/enhance the robustness of LLMs using KGs. The work was conceived, developed, and validated considering the political domain. The text is very clear, and easy to follow. The presented method is novel, and I believe it is aligned with this conference's thematic.

Despite the chosen domain, I believe the method could be generic, and applicable to several other domains. I would suggest the authors, in case they should further enhance it in a forthcoming study, to present it as so, and consider the political domain as a case study, along with other domains.

**Questions:**

- Did you evaluate the results of directly incorporating the data/knowledge embedded in KGs into the LLM itself? Are the answers obtained from such an approach the same as using your method? (this could be part of a future work)
- What would be the necessary steps to apply your method to different domains, other than Politics? What would be the size of this gap? This could be a relevant discussion for extending your study.
- Do you have a closed T-Box in your local KGs? I see in the figures/examples a specific set of relationships, is this required for your method to work, or could it also work with distinct relationship types?
- Would enriching your local KGs with further/derived relationships be applicable? Could this possibly further improve the responses at the end of the overall process? (this could be part of a future work)

**Reviewer Confidence:**

2: The reviewer is willing to defend the evaluation, but it is likely that the reviewer did not understand parts of the paper

**Scope:**

4: The work is relevant to the Web and to the track, and is of broad interest to the community

---

### Official Review · Reviewer_DyjY · 2023-11-26

**Novelty:** 4
**Technical Quality:** 5

**Review:**

The paper addresses the critical challenge of incomplete and outdated intrinsic knowledge in the political domain by proposing the integration of a domain-specific political knowledge graph. This innovative solution holds the promise of significantly enhancing the performance of Language Models (LLMs) in political tasks. The conducted experiments across diverse datasets and LLMs compellingly demonstrate the efficacy and explainability of the proposed Political Experts through Knowledge Graph Integration (PEG) framework, marking a meaningful contribution at the intersection of language models and political expertise.

The writing is generally clear, and the method is straightforward, focusing on grouping and retrieving relevant knowledge triplets for precise LLM generation. The paper's notable contribution relies on the construction of the dataset knowledge graph, while the LLM inference part might benefit from a more in-depth ablation study to identify optimal settings for improved prompting.

While the overall structure is well-organized, certain areas could be refined for better clarity. Notably, an explicit explanation of Table 1 is absent, leaving readers to interpret the significance of the provided information. Additionally, the figure illustrating the functioning of the prompting system remains somewhat challenging to grasp without a clear caption, with a more comprehensive understanding gained from the appendix. Addressing these aspects will undoubtedly enhance the overall clarity of the paper.

Additionally, the paper introduces the concept of hallucination but lacks an in-depth discussion of how the proposed method addresses this aspect in the experimental section.


Pros:

- The integration of a domain-specific political knowledge graph provides an effective solution to address knowledge gaps in LLMs.
- The PEG framework successfully unifies local and global knowledge, demonstrating both effectiveness and explainability.
- Inclusion of experiments across different datasets and LLMs strengthens the proposed approach.

Cons:
- The method makes a valuable contribution to knowledge acquisition but may have limitations in terms of novelty in the inference or evaluation part.
- Prompting Optimization: It remains somewhat unclear how the best prompting is determined, suggesting the need for further clarification.
- Writing Refinement: The lack of an explicit explanation for Table 1 and the somewhat unclear figure depicting the prompting system could be areas for improvement.
- Terminology Clarification: The term "global vector" requires a more precise definition to avoid confusion regarding its role in the presented figure.
-  The paper does not use \citet{} as the appropriate citation format.

**Questions:**

Given the focus on binary classification tasks, have you considered including a supervised learning baseline with smaller models that incorporate the constructed knowledge graph? How might this contribute to the evaluation and interpretation of results?

**Reviewer Confidence:**

3: The reviewer is confident but not certain that the evaluation is correct

**Scope:**

3: The work is somewhat relevant to the Web and to the track, and is of narrow interest to a sub-community

---

### Official Review · Reviewer_rApu · 2023-11-27

**Novelty:** 4
**Technical Quality:** 4

**Review:**

This paper constructs a domain-specific political knowledge graph covering diverse facets of U.S. politics. The Political Experts through Knowledge Graph Integration (PEG) framework is proposed to address the tasks of political actor and opinion mining.
Then it unifies the local and global knowledge using diverse methods, to alleviate the issue when direct answers can not be found in constructed knowledge graphs. Experiments show the effectiveness of the proposed model.

I give a weak acceptance of this paper, as it generally proposes a domain-specific  LLM for political actor modeling and opinion mining.
Strengths:
a) it is a well-structured and easy-to-follow paper.
b) the proposed method seems interesting.

Weakness:
Experiments are not convincing enough. The running and inference efficiency analyses of the model are expected to be shown. The paper has few comparable baselines. For example, "Baselines without External Knowledge" only includes two models, the GPT and other latest LLM models are expected to be compared. In addition, The experiment results lack variance, casting doubt on the confidence of the results.

**Questions:**

a) The running and inference efficiency analyses of the model are expected to be shown.
b) Few comparable baselines. For example, "Baselines without External Knowledge" only includes two models, the GPT and other latest LLM models are expected to be compared.
c) The paper does not provide the source codes and the reproducibility of the experiment is uncertain.
d) The experiment results lack variance, casting doubt on the confidence of the results.

**Reviewer Confidence:**

3: The reviewer is confident but not certain that the evaluation is correct

**Scope:**

3: The work is somewhat relevant to the Web and to the track, and is of narrow interest to a sub-community

---

### Official Review · Reviewer_vSdb · 2023-11-30

**Novelty:** 6
**Technical Quality:** 6

**Review:**

The author constructs a political knowledge graph MVPKG and utilizes it to empower LLM in political tasks. The process of knowledge utilization includes knowledge acquisition, aggregation, and utilization.
For knowledge aggregation, the paper proposes implicit aggregation with embedding techniques and explicit aggregation with natural language. Global knowledge is introduced to enhance the effectiveness of political tasks and alleviate the lack of political knowledge in large models.

pros:
The method is clearly delineated and readily comprehensible, facilitating replication and understanding by other researchers. The proposed empowerment of LLMs using a knowledge graph is innovative and of significant interest to peers in the field.
The experiments conducted across three distinct datasets robustly substantiate the efficacy of the proposed method. The analysis of the experimental results is thorough and comprehensive, providing a detailed examination of the method’s impact.

weakness:
In explicit eggregation, the prompt seems naive and addition tune may provide better results.

In conclusion, the paper offers a significant contribution by demonstrating how a political knowledge graph can enhance the capabilities of LLMs within the political sphere.

**Questions:**

1. In explicit eggregation, the prompt seems naive and addition tune may provide better results.

**Reviewer Confidence:**

3: The reviewer is confident but not certain that the evaluation is correct

**Scope:**

3: The work is somewhat relevant to the Web and to the track, and is of narrow interest to a sub-community

---

### Decision · Program_Chairs · 2024-01-22

**Decision:**

Accept (Oral)

**Comment:**

The reviewers give consistently high scores.
 Although several points were raised (see below), I consider the paper strong enough for acceptance.
 Some reviewers proposed to do a more in-depth investigation of optimal prompts and of runtime complexity, but in these case these aspects don't form the core contribution, so I don't consider them show-stoppers.
 One reviewer also proposed that the framework could be applied more broadly than politics, but as discussed this might be more appropriate for a future paper.